

# Landslides Data Assimilation Using TRIGRS Based on Particle Filtering

Changhu Xue[1], Guigen Nie[1,2], Jie Dong[3], Shuguang Wu[1], Jing Wang[1], Xiuzhen Li[4], Xiaogang Zhang[4]

[1]GNSS Research Center, Wuhan University, Wuhan, 430079, China
[2]Collaborative Innovation Center for Geospatial Information Technology, Wuhan, 430206, China
[3]State Key Laboratory of Information Engineering in Surveying, Mapping and Remote Sensing, Wuhan University, Wuhan, 430079, China
[4]Institute of Mountain Hazards and Environment, Chinese Academy of Sciences, Chengdu, 610041, China

*Correspondence to: Guigen Nie (ggnie@whu.edu.cn)*

**Abstract.** Studies about landslide modeling and monitoring are becoming more diverse. Data assimilation is an approach to combine mechanism models and observations. In this study, an improved particle filtering algorithm is used to assimilate the transient rainfall infiltration and grid-based regional slope-stability analysis (TRIGRS) model and landslide surface deformation monitoring data observed with GPS and InSAR. After assimilation calculation, FS has been effectively corrected, rather than continuously decreasing as the background model output. The root mean square difference (RMSD) tends to decrease from a maximum of 0.084 to a minimum of 0.026 in the process of assimilation, which means the assimilation process makes the model output FS closer to the actual observations. The friction angle ($\varphi$), which is an investigated parameter, can be updated and fed back in each step of assimilation. The value of the investigated parameter makes the model output closer to the observation. The groundwater pressure head is output as an assimilation result simultaneously.

## 1 Introduction

Landslides induced by rainfall pose a huge threat on human lives and properties around the world(Hungr, et al., 2001, Kirschbaum, et al., 2010). Currently, there are varies types of methods for landslides analysis, such as physical mechanism modeling(Wu and Sidle, 1995, Iverson, 2000, Hong, et al., 2007, Baum, et al., 2008, De Blasio and Crosta, 2017, Martelloni, et al., 2017) , 3D modeling based on numerical analysis(Crosta, et al., 2003, McDougall and Hungr, 2004, Hungr and McDougall, 2009, Merritt, et al., 2014, Yang, et al., 2014, Martelloni, et al., 2017) and time series analysis of landslide deformation (Gokceoglu and Aksoy, 1996, Hong, et al., 2007, Rossi, et al., 2010, Liu, et al., 2013, Turner, et al., 2015, Dong, et al., 2018). The transient rainfall infiltration and grid-based regional slope-stability analysis (TRIGRS) model program coded in Fortran computes transient pore-pressure changes, and attendant changes in the factor of safety (FS), due to rainfall infiltration. It is designed for modeling the timing and distribution of shallow, rainfall-induced landslides(Baum, et al., 2008). Some studies about landslides safety analysis using TRIGRS are developed in recent years(Liao, et al., 2011, Park, et al., 2013, Bordoni, et al., 2015, Viet, et al., 2017). Increasing studies about landslide surface deformation monitoring and analysis are



carried out, including GNSS, SAR/InSAR technology, three-dimensional laser scanning(Squarzoni, et al., 2005, Brueckl, et al., 2006, Du and Teng, 2007, Peyret, et al., 2008, Yin, et al., 2010, Dong, et al., 2018).

Data assimilation (DA) is a method of combining a dynamic system and observations of its states, in order to improve the accurate description of the system, including its uncertainty analysis. It uses all available observed information and background model to simulate the real process, estimate uncertain state and parameters, and thus improve the prediction(Talagrand, 1997, Evensen, 2009). DA is widely applied in many fields such as oceanic and atmospheric modeling for prediction in different scales, regional or global hydrology research including parameter estimation and dynamic analysis, and evolution and inversion of land surface process(Kalnay, et al., 1996, Houtekamer and Mitchell, 2001, Rowley, et al., 2002, Uppala, et al., 2005, Salamon and Feyen, 2009, Plaza, et al., 2012, Mazzoleni, et al., 2018). But in the scope of landslide research, there are only a few preliminary studies(Jiang, et al., 2016, Xue, et al., 2018). Using the TRIGRS program as the background, we studied the approach to merging observations of landslide surface deformation into the evolutionary model, and investigated the update and feedback of the parameter friction angle (φ).

A great number of approaches of DA have been developed in recent years, of which sequential algorithms like particle filtering (PF) are increasingly popular. PF is based on Bayesian theory and originally introduced by Arulampalam into DA(Arulampalam, et al., 2002). It is developed in many DA studies due to the advantage of being unconstrained by state Gaussian distribution and linear assumptions(Weerts and El Serafy, 2006, Chorin, et al., 2010, Liu, et al., 2013, Thirel, et al., 2013, Fearnhead and Kunsch, 2018). However, PF still has disadvantages like particle degradation, which means with the recursion of the system most particles' weights tend to be zero and only a few particles have effects on the results(Carpenter, et al., 1999). Therefore, a number of improved algorithms for particle filtering have been proposed(Pitt and Shephard, 1999, Kotecha and Djuric, 2003, Khan, et al., 2005, van Leeuwen, 2010, Zhang, et al., 2013, Wu, et al., 2014, Xi, et al., 2015).

In this study, an improved particle filter algorithm(Xue, et al., 2018) is adopted to perform assimilation experiments on TRIGRS landslide models and deformation observation data in the area of Xishancun landslide, Sichuan Province, China. InSAR and GPS monitoring data in the study area are chosen as observations of assimilation. Taking friction angle (φ) for example, we investigate the sensitivity analysis to determine the original value of a parameter and propose the parameter update and feedback mechanism to adapt the model to observations. The assimilation results and the change trend of friction angle during the assimilation period are illustrated in the paper. Results show that the assimilation can effectively improve the accuracy of state estimation by fixing the TRIGRS model to be close to observations.

## 2 Geologic background

The study area is located at around 31°35'N,103°26.5'E (Figure 1), the northern bank of Zagunao River in Xishancun, between Wenchuan and Li County, Sichuan, China. It is at the east of the Tibetan Plateau, where exists a strong crustal activity and influenced by the Wenchuan earthquake in 2008. The front and rear boundaries are respectively 1510m and 3300m above





the sea level. The whole length of the landslide is about 3800m, and the width is varied in a range of 680m to 980m(Qu, et al., 2016).

Xishancun Landslide is divided into three parts according to different soil conditions as Figure 2 shows. The slope of the lower block is between $27°$ to $32°$, and the average slope of the middle block and upper block are $22°$ and $27°$ respectively. The landslide body is the northern side of a V-shaped valley which is inhabited by a number of villagers, and slides from north to south. Its southern edge is close to Wenma Expressway and National Highway 317. Affected by human activities, such as terraces reclamation and road construction, some areas form severe steep slopes or collapses.

## 3 Data observation and processing

### 3.1 Collection of Observations

On the surface of Xishancun Landslide, 4 GPS monitoring station are receiving positioning observations along the sliding direction of the landslide body. One base station is established on the bedrock of the adjacent mountain. Figure 3 shows the positions of GPS stations on the slope. GPS Monitoring data is collected from Aug 12, 2015, to Nov 07, 2017, one set of observations every 10 days. In order to eliminate the influence of plate motion, we use the relative positioning method to calculate the displacement of each monitoring station related to the base station (Figure 4).

Another data set is InSAR observations of Sentinel-1 interferometric wide (Sentinel-1 IW) data. The Sentinel-1 IW image width is 250km, and each image consists of three left and right overlapping frames. We collect the Sentinel-1 IW data for the period of Aug 5, 2015, to Jul 13, 2017, a total of 34 sets of data. Most of these data sets are separated by 12 or 24 days and a few by 6 days. An average velocity map of deformations per-year is illustrated in figure 5. However, the displacement of InSAR observation is always along the direction of the radar line of sight (LOS). In order to verify the consistency of GPS and InSAR observations, these two observations must be projected in the same direction. The conversion vector of GPS 3D-displacement projection to the line of sight is $\begin{bmatrix} -0.13 & 0.58 & 0.80 \end{bmatrix}$. InSAR observations have been corrected for atmospheric delays to reduce fluctuations. We select the displacement of the InSAR observation with a radius of 50 meters around the GPS station and compare it with the GPS observing results. Sequences of corrected InSAR and projected GPS observations are displayed in figure 6. As we can see, the two observing results are consistent in both trend and quantity in most circumstances.

According to the post-failure movement model of a landslide triggered by rainfall infiltration, the relationship between FS and displacement rate is expressed as

$$\frac{1}{g}\frac{dv}{dt} = \sin\alpha[1 - FS] \tag{1}$$

where g is the magnitude of gravity acceleration, α is the slope angle(Iverson, 2000). The observed velocities can be converted into FS using formula (1) when the slope is slipping. Therefore, the converted FS can be used as a set of observations for sensitivity analysis and assimilation experiment.





### 3.2 Sensitivity analysis

The initial value and trend of some parameters in landslide evolution are usually difficult to exactly determine. In this work, the friction angle ($\varphi$) is selected to conduct the sensitivity analysis. Other parameters are fixed while the friction angle is adjusted from minimum to maximum. The root means square difference (RMSD) of all grid cells is used as the evaluation of sensitivity analysis.

$$RMSD_k = \sqrt{\frac{1}{N_p} \sum_{i,j} (FS_{ij}(\varphi_k) - FS_{ij}^{obs})^2} \qquad (2)$$

where $FS(\varphi)$ is the FS calculated by TRIGRS program using parameter $\varphi$, $FS^{obs}$ is the converted FS from GPS/InSAR observations, $N_p$ is the total number of grid points, $i, j$ are row and column number, respectively. Figure 7 shows the RMSD sequences of the 3 landslide blocks calculated by sensitivity analysis of $\varphi$. The result shows that the model is best performed when initial friction angles of 3 blocks are $19.2°$, $22.0°$, $22.2°$, respectively.

### 3.3 Assimilation experiment

In order to facilitate the realization of the experiment, the FS is used as an assimilation variable for one-dimensional data assimilation experiments. The assimilation experiment starts on the first day of observation data and occurs daily using the improved PF. If there is observation data on the current day, the assimilation is performed, otherwise, only the model recursion is performed. The friction angle is updated at the same time of each assimilating calculation with formula (3).

$$\hat{\varphi}_k = \sum_{i=1}^{N} \varphi_k^i \cdot w_k^i \qquad (3)$$

in which $\hat{\varphi}_k$ is the estimation of $\varphi$ at step k, $\varphi_k^i$ is the $i$th sample of $\varphi$ in particle filtering, $w^i$ is the corresponding sample weight.

The FS results of background model and assimilation output are illustrated in figure 8. Compared with the model output, there is a significant change in the safety factor after assimilation. In the background of the TRIGRS program, the only changing parameter is rainfall, so the model output results have small fluctuations and gently decreases with time. Assimilation results are more dependent on observations.

Figure 9 shows sequences of model output and assimilated results of three grid cells randomly selected from Block I, II, III, respectively. It can be seen that the change tracks of FS are corrected by observations obviously. Since rainfall is the only variable input parameter in the background model, the TRIGRS output FS sequences present relatively stable downward trends. Although significant fluctuations are obvious in observations and are propagated into assimilation results, the assimilation series has been significantly improved. The RMSD sequence of assimilation output FS on the whole landslide calculated by the formula (2) is shown in figure 10. Its maximum value is 0.084 at the second step of assimilation, and the minimum is 0.026



in the last several steps. Overall, with the assimilation proceeding, the RMSD of results tends to become smaller and its sequence gradually becomes stable.

The friction angle ($\varphi$) is a parameter to be investigated in this experiment, so the distribution and the changes are displayed in figure 11 and time series of the 3 grid cells are shown in figure 12. In the assimilation progresses, the change of internal friction

angle is mainly affected by the actual soil water content and deviation of other initial parameters.

Convert the assimilated FS into annual average deformation rate using formula (1), its distribution map is as figure 13 shows. Groundwater pressure head ($\psi$) is an important parameter in slope stability analysis. In this experiment, $\psi$ is also considered as an output parameter. Figure 14 demonstrates the distribution of $\psi$ and its change with assimilation time. The $\psi$ time series of the above 3 points is illustrated in figure 15. It can provide a reference for slope analysis.

**4 Conclusion and discussion**

Data assimilation is a method that can combine mechanism models with observational data. In data assimilation fields, particle filtering is becoming increasingly popular since it can help to solve non-linear and non-Gaussian problems efficiently. In this study, data assimilation with an improved particle filtering algorithm is applied to landslide model and observation data processing. TRIGRS model is used as the background, GPS and InSAR monitoring data as input observation in the experiment.

Results suggest that the fs sequence of TRIGRS output decreases continuously with time and the assimilation can effectively correct the FS of the model output so that it does not deviate too much from the actual. The RMSD of FS indicates the assimilation results can correct the estimation of TRIGRS output close to actual observations. The experiment also examined the changes of soil friction angle and the groundwater pressure head which can help to analyse the soil water content and slope stability in the landslide body.

This paper provides an approach to apply data assimilation method to stability analysis and parameter update and feedback in a landslide. Landslide data assimilation experiments have many directions for further research, such as better mathematical models, more comprehensive and high-precision observations, and more excellent assimilation algorithms.

*Author contributions.* CX conceived the idea of this article and completed the paper. GN helped to solve some important problems and proposed some key suggestions. JD provided the data analysis of GPS/InSAR. SW and JW made contributions
to the validation of the algorithm, chart sorting and editing. XL and XZ helped to build the model with TRIGRS program.

*Competing interests.* The authors declare that they have no conflict of interest.

*Acknowledgments.* This work is financially supported by the National Key Basic Research Program of China (grant no. 2013CB733205).



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

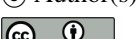


**Figures**

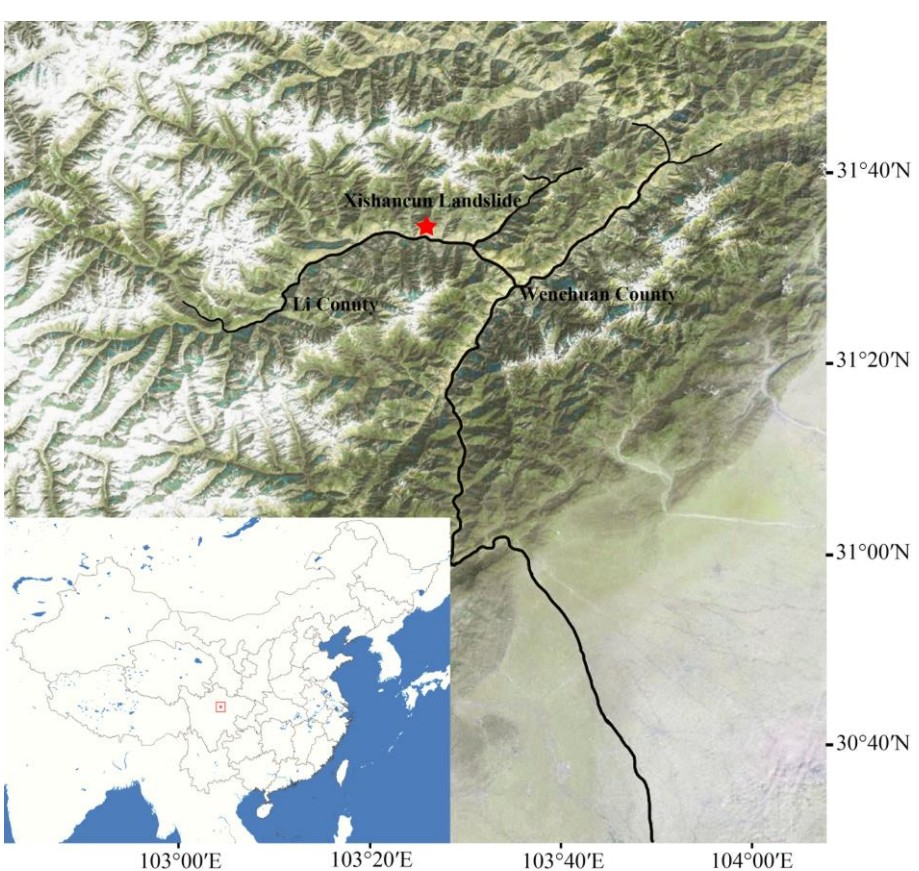

**Figure 1.** Location of Xishancun Landslide in the red rectangle and its location in China.



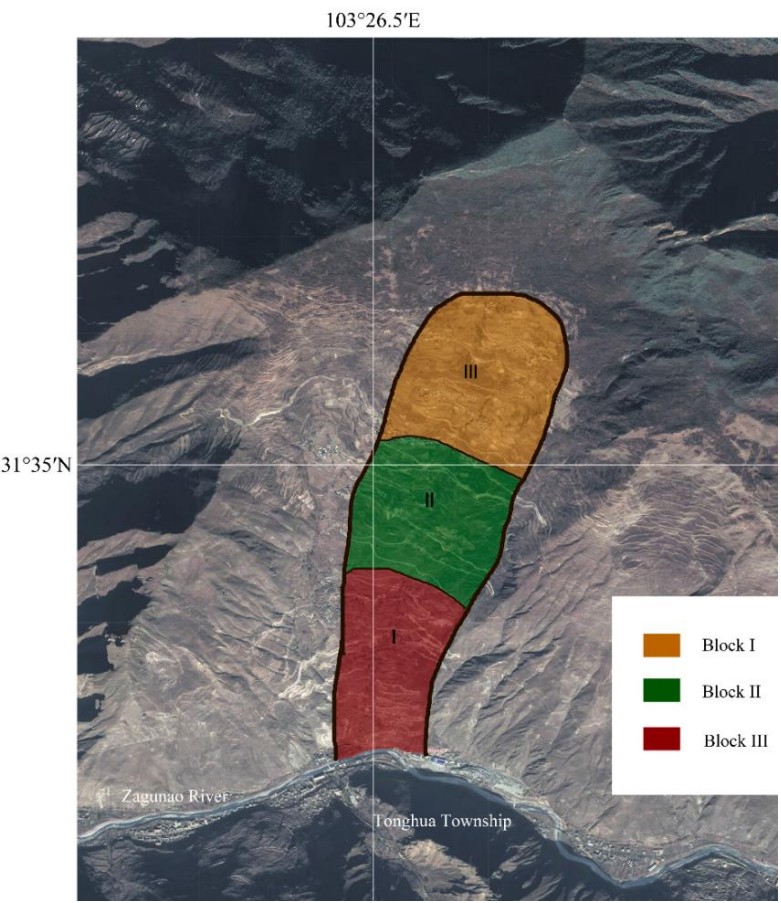

**Figure 2.** The top view and blocks of the study area.





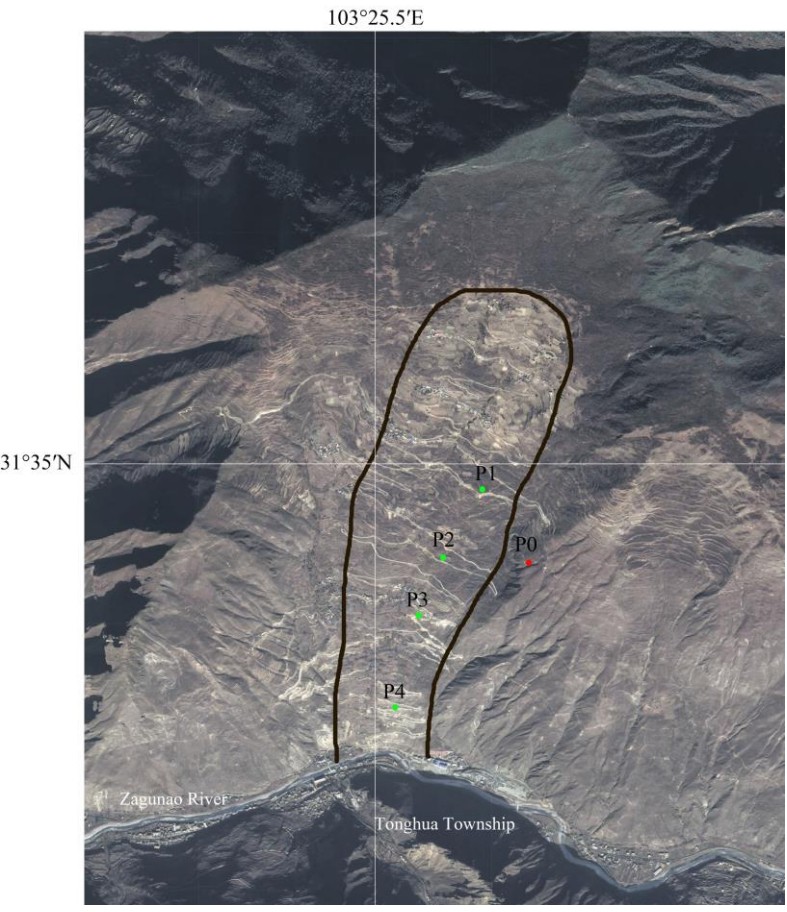

**Figure 3.** Location map of GPS monitoring station. P1-P4 are monitoring stations on the landslide surface, P0 is the base station built on bedrock.





**Figure 4.** Displacement series of the 4 GPS monitoring stations. (a), (b) and (c) are the figures of displacements in the north, east and upper directions respectively, (d) is the total displacements figure.



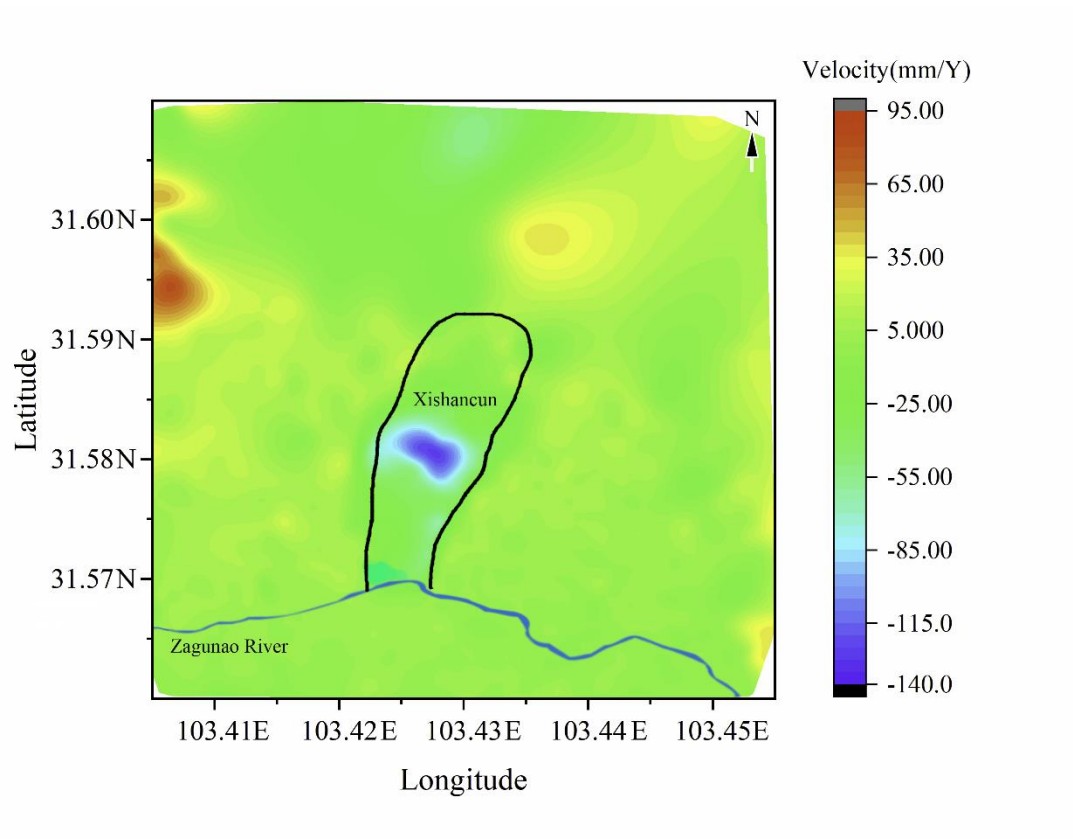

**Figure 5.** Displacement rate map of Xishancun area monitored by Sentinel-1 data.



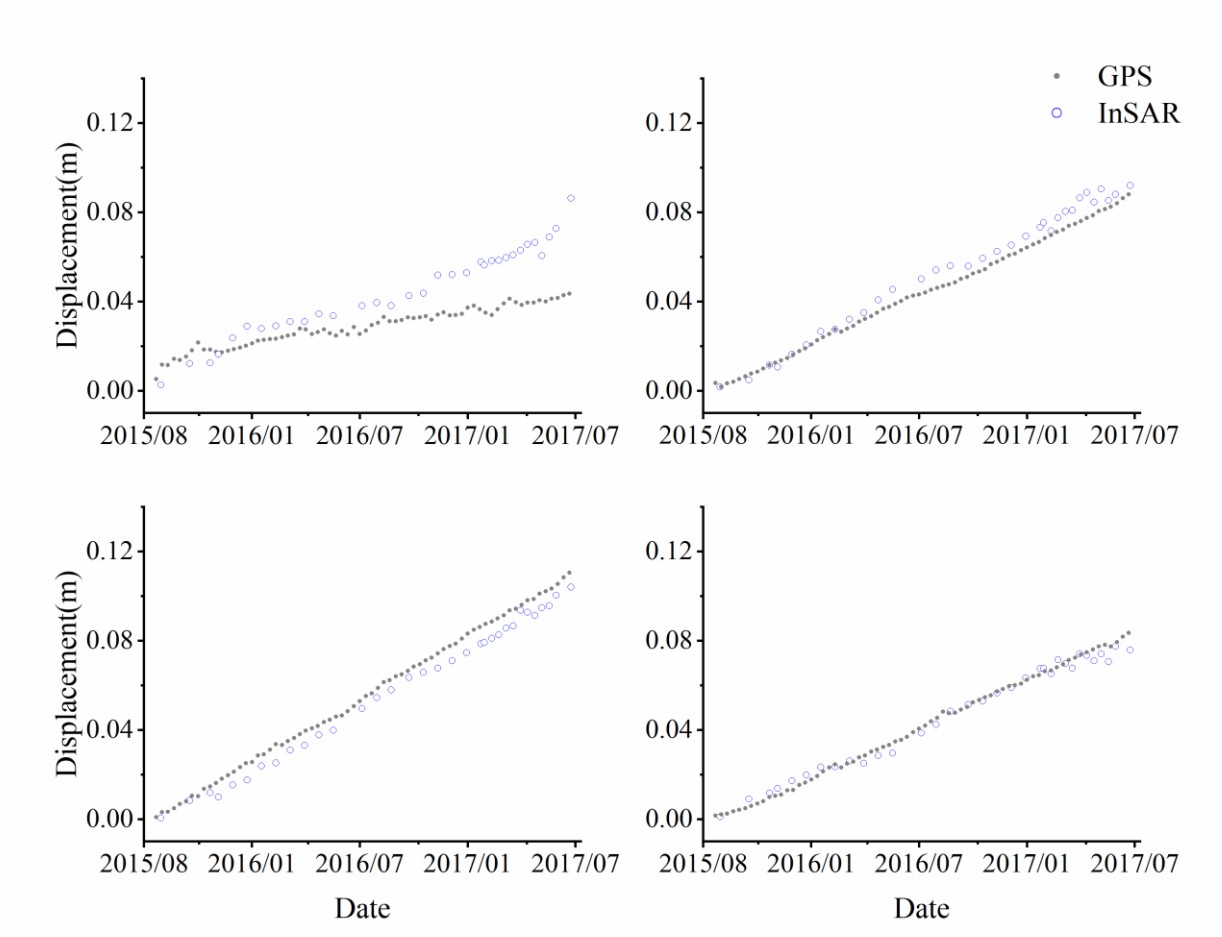

**Figure 6.** Comparison of GPS and INSAR monitoring displacement sequences in LOS direction.



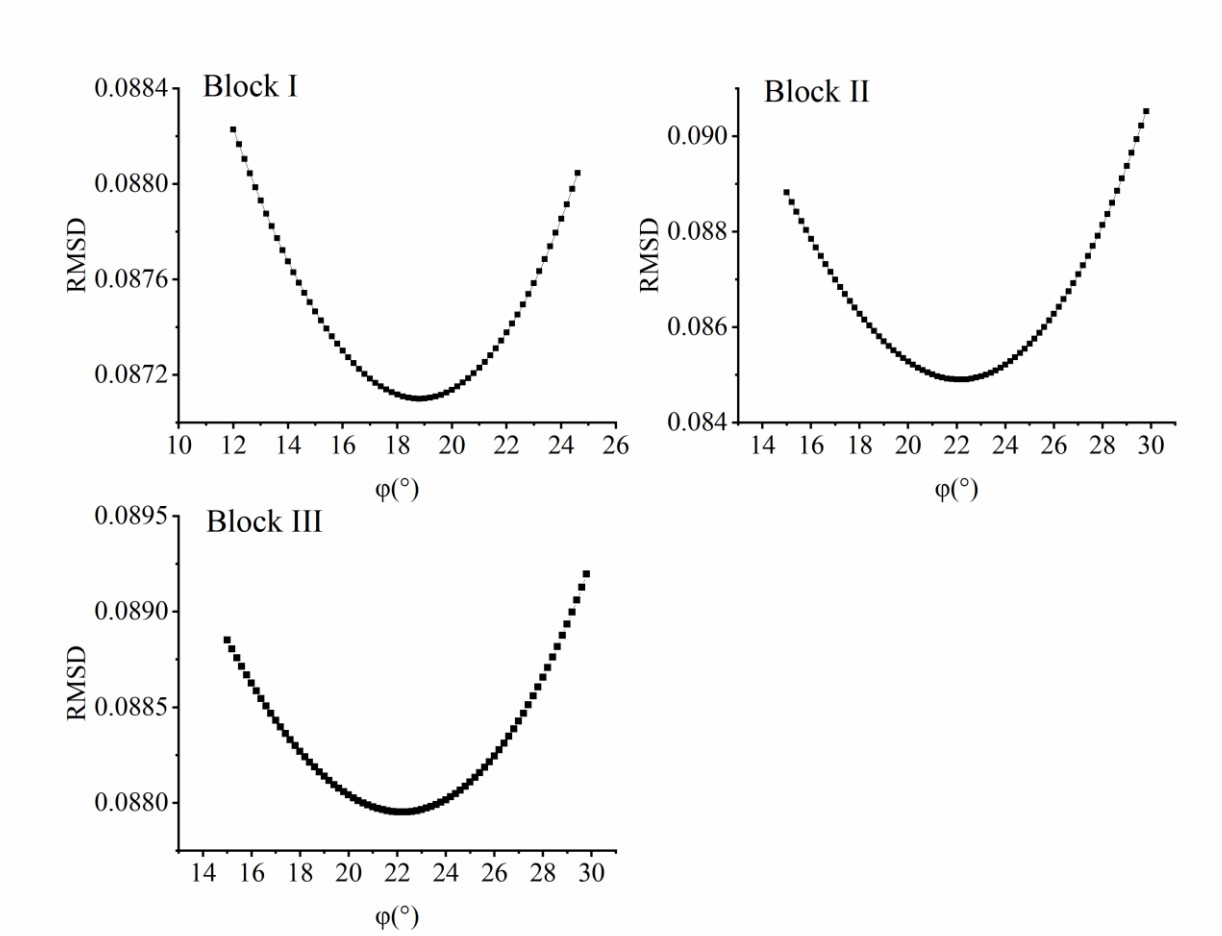

**Figure 7.** Sensitivity sequences of RMSD of 3 blocks changing with φ.




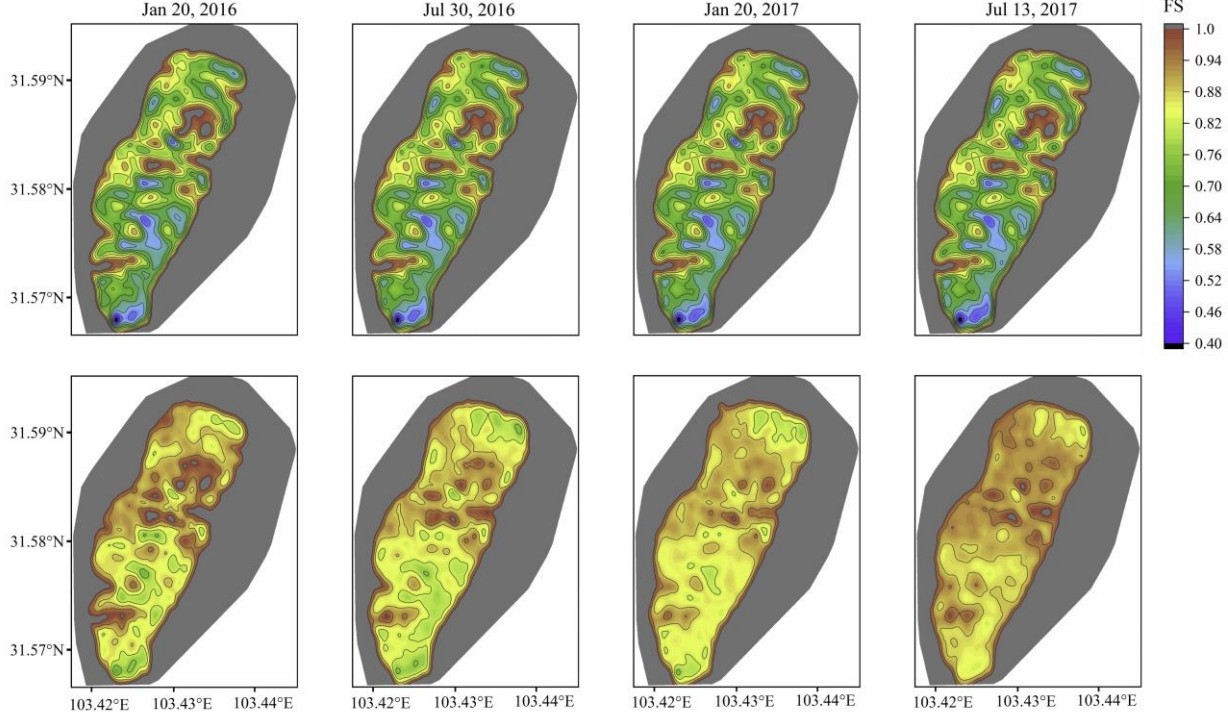

**Figure 8.** Comparison of background model output results and assimilation results. The maps in the first line are TRIGRS output results, and those in the second line are assimilation results.



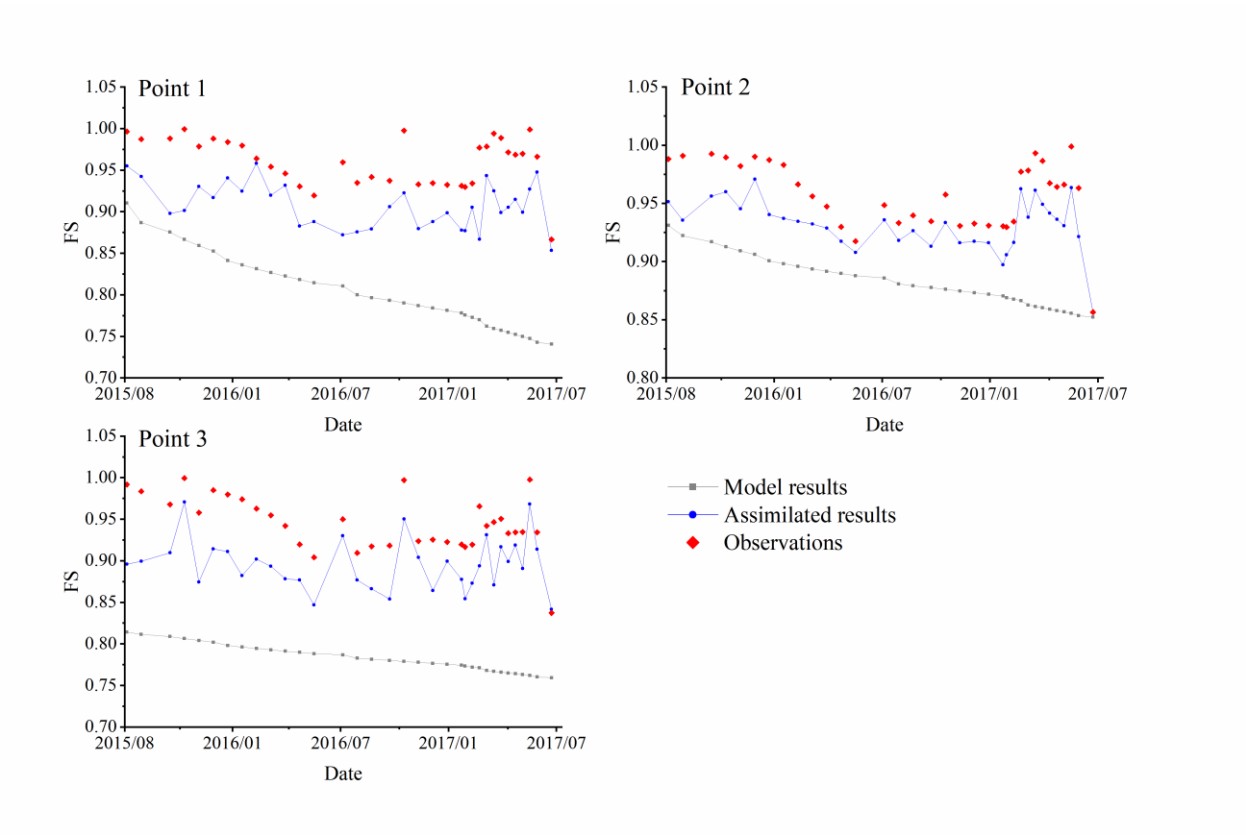

**Figure 9.** Model and assimilation sequences of 3 points selected from Block I, II, III, respectively.



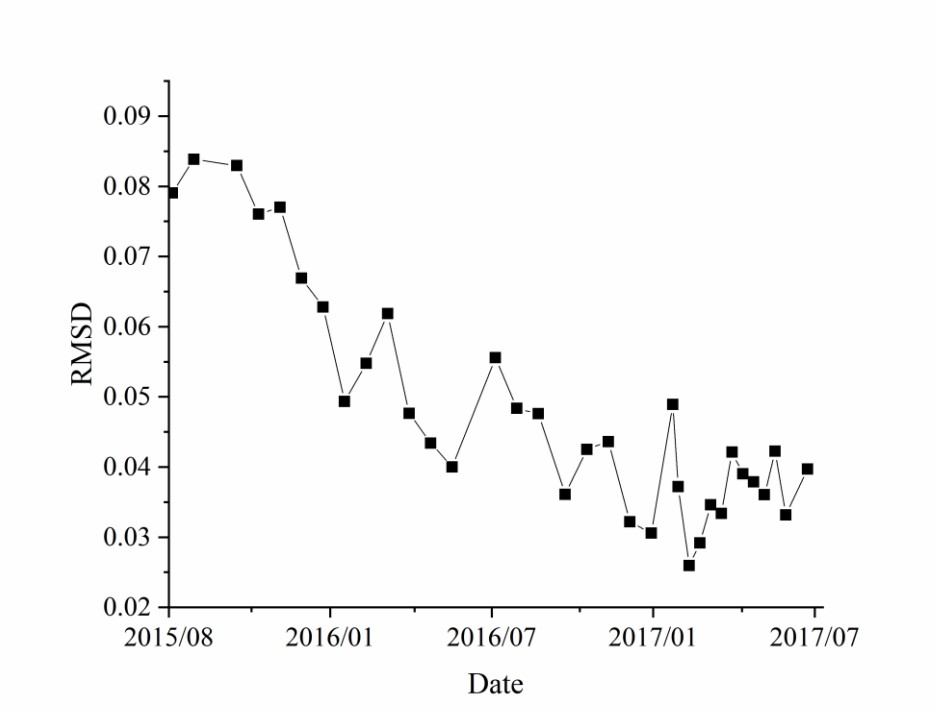

**Figure 10.** The RMSD sequence of assimilation output FS related on observations.





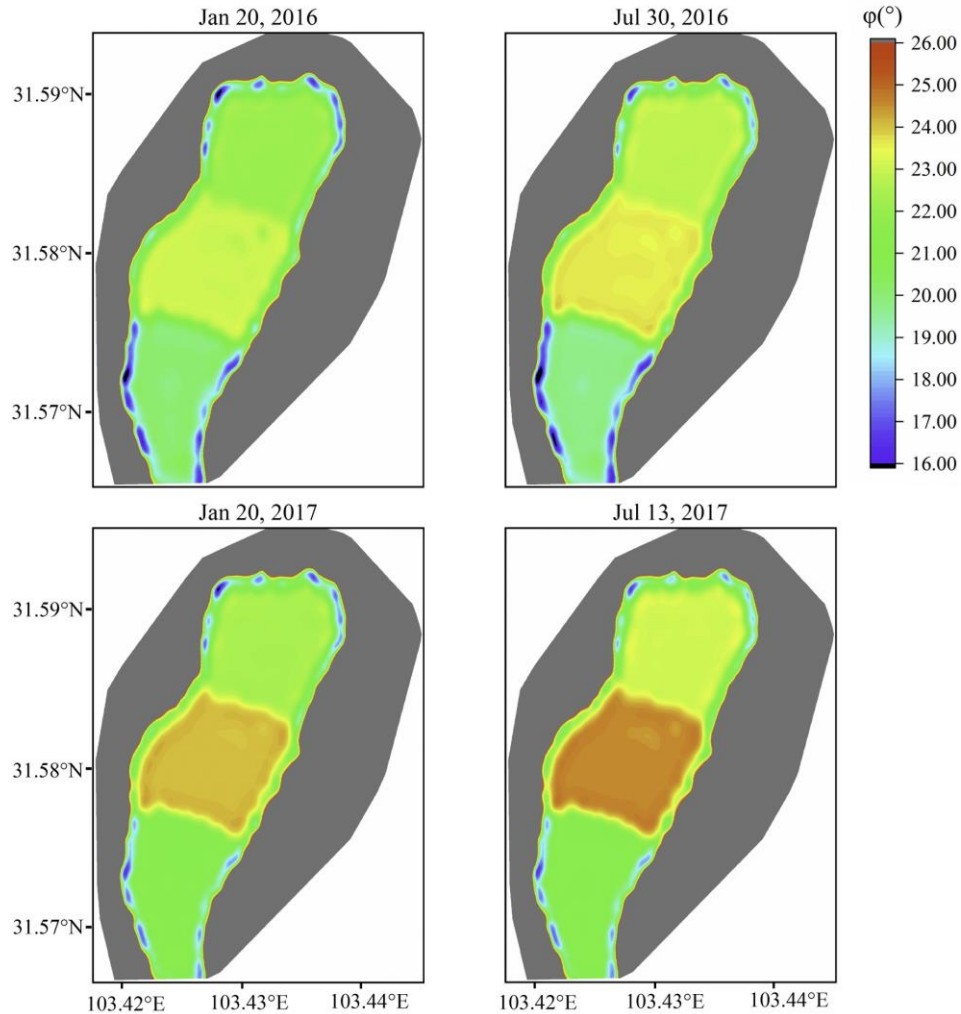

**Figure 11.** The distribution of friction angle and its change with assimilation time. There are obvious boundaries in the

distribution map because the TRIGRS program divides different blocks into zones.



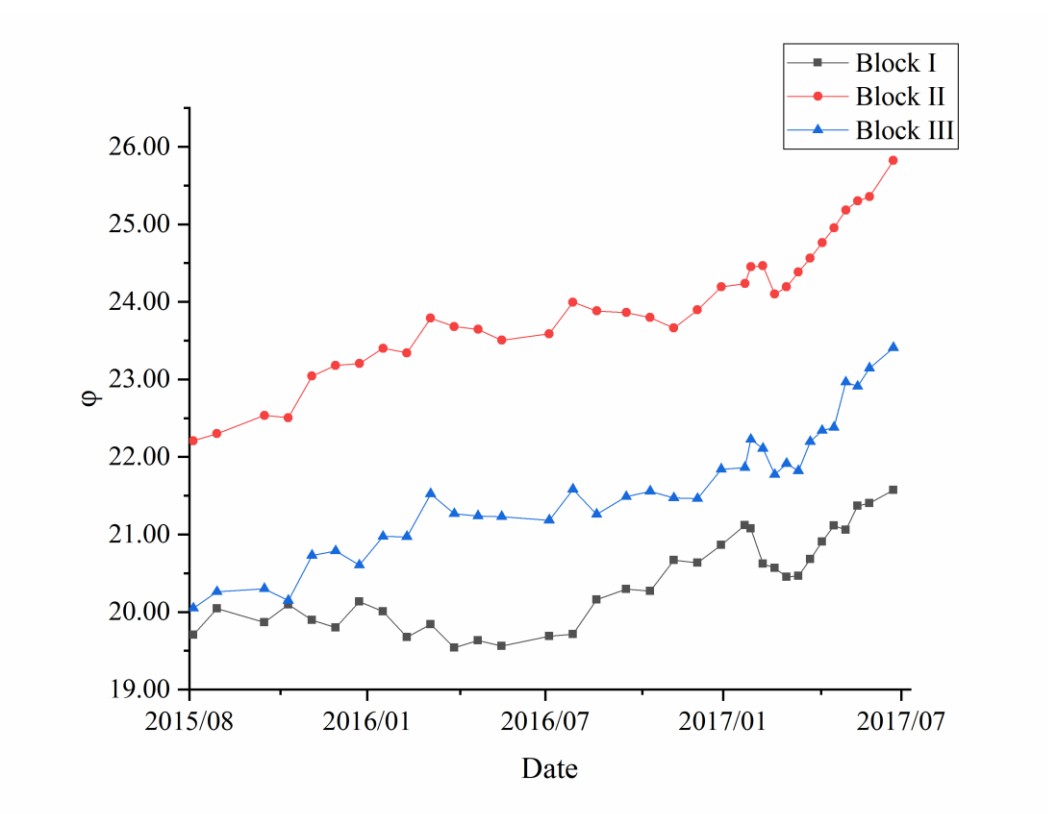

**Figure 12.** Friction angle time series of three points after assimilation.



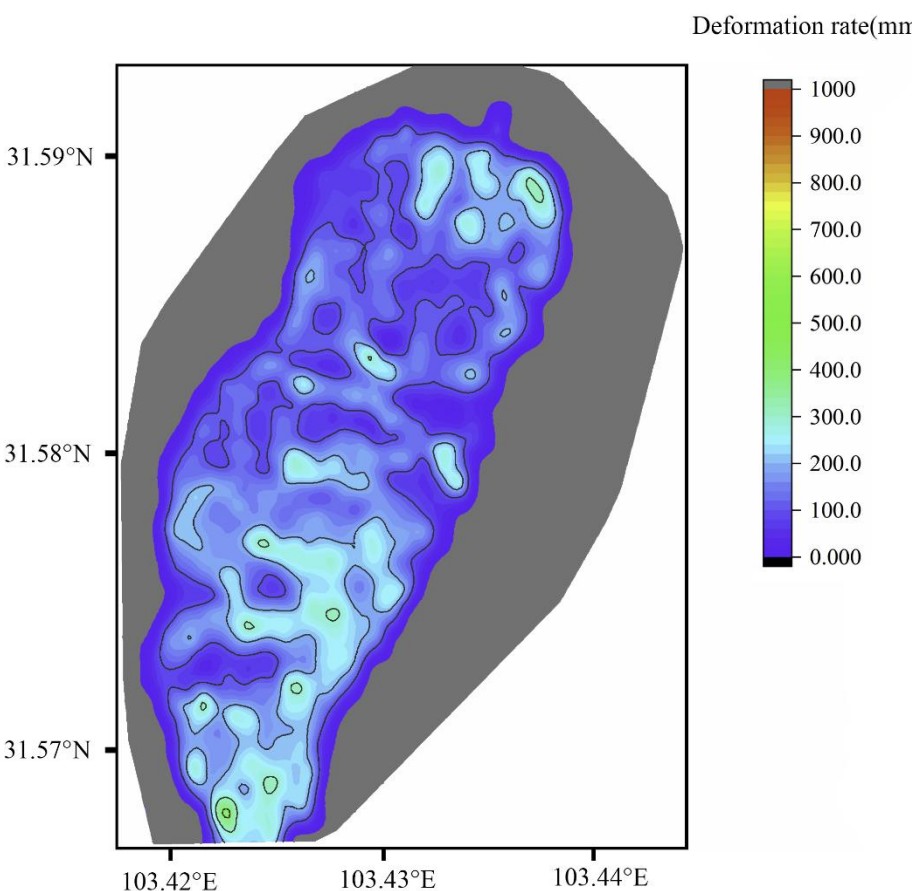

**Figure 13.** Distribution of annual deformation rate calculated by the assimilated FS.



**Figure 14.** The distribution of groundwater pressure head and its change with time.





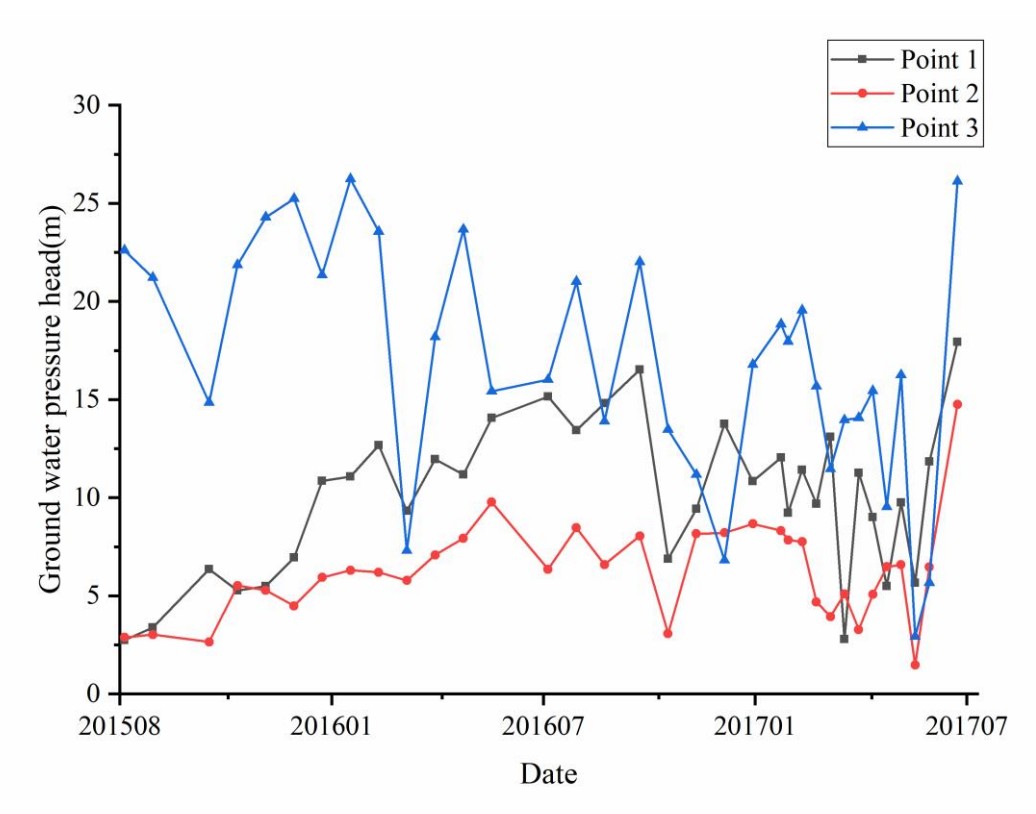

**Figure 15.** The groundwater pressure head time series of the three points.