# Peer review of "Landslides Data Assimilation Using TRIGRS Based on Particle Filtering"

_Natural Hazards and Earth System Sciences, 2019_

## Referee Comment (RC1) · Anonymous Referee #1 · 26 Feb 2019

General comments: The paper is an interesting application of data assimilation to time-series modeling of a slow landslide using TRIGRS. The use of TRIGRS in this respect is novel to my knowledge. The paper is well-organized, however I believe the discussion is lacking sufficient detail and explanation which would aid the understanding of the reader and enhance the impact. In particular, the ultimate added value of this procedure is not well-defined. If calibrated model inputs are desired, it makes sense to discuss the various other interdependent input parameters of TRIGRS. If corrected model outputs are desired, then more discussion of how the corrected TRIGRS outputs compare to observations and to uncorrected outputs is needed. The figures have a lot of data on this subject but discussion of the figures is too brief in my opinion.

Specific comments:

[Figure]

1. pg2 line 4, the word "background" is used throughout the manuscript and is clearly used in reference to the TRIGRS model, but the meaning isn't quite clear to me. If this is a term more common to DA literature it would help to more explicitly define it here.

2. pg 2 line 10, another recent DA application to natural hazards is Brezzi et al. (2016). "A new data assimilation procedure to develop a debris flow run-out model" Landslides. October 2016, Volume 13, Issue 5, pp 1083–1096 https://doi.org/10.1007/s10346-015-0625-y

3. pg 2 line 30, "It is at the east of the Tibetan Plateau,..." This sentence is awkward and should be reworded.

4. pg 3 line 25, the projected GPS and InSAR tend to agree, but in the DA procedure, which is used for comparing FS values? InSAR only? Is the GPS essentially a check on the InSAR?

5. pg 4 line 3, "Other parameters are fixed." Please mention how other parameters are determined, especially rainfall intensity, and if their sensitivity affects friction angle. Conductivity and other hydraulic parameters will have a significant effect on the time-evolution of output FS and directly relate to the pressure head output. Why was friction angle chosen? Please explain this choice and the choice of other parameters. If the purpose of DA is to correct model outputs as opposed to back-calculating "true" model inputs, then I can see why it wouldn't be as important to calibrate all the input parameters. If that is the case, I think a bit more discussion is needed highlighting the improvement and emphasizing the utility of the corrected outputs.

6. pg 4 line 7, is the sensitivity analysis performed at the first time step? Please clarify.

7. pg 5 line 5, "change of internal friction angle is mainly affected by soil water and deviation of other initial parameters." I'm not clear on what is meant by this. I'm guessing it means that changing friction angle is effectively correcting for inaccuracies in the hydraulic parameters. But the pressure head output is independent of friction angle,

so I'm having trouble understanding the added value of DA with regard to the pressure head outputs. Regardless, I think this sentence should be elaborated on.

Regarding Figures 9 and 12: If TRIGRS FS continues to decrease while the observed velocity stays constant, then would friction angle continue increasing arbitrarily until it would eventually reach unrealistic values?

8. pg 5 line 6, how does the calculated average deformation rate compare to the observed?

9. pg 5 line 9, "It can provide a reference for slope analysis." Again, I think it would be good to expand on how the corrected outputs are useful and explain why others might find utility in replicating this procedure.

Grammar and figures: The English of this paper is good, however there are many instances of slightly awkward wording, a few misspellings, and inconsistent capitalization.

Figures 1, 2, and 3 should be improved. Text size should be increased and some sort of contrast should be added to make it more readable against the dark background. The inset map on Figure 1 needs to better identify the study area location (the red box is too small to see clearly). All maps require a scale bar and north arrow.

---

## Referee Comment (RC2) · Anonymous Referee #2 · 14 Mar 2019

Summary: Landslide stability modelling is conducted using a number of methods, including the USGS-built TRIGRS system. Numerical models of this sort have been in some instances improved by using 'data assimilation' methods, meaning, as I understand it, reinitializing model parameters during model runs with observational data to better represent reality. Here the authors describe the use of a data assimilation method to improve the performance of the TRIGRS model on a specific test case in Central China. They suggest that the model outputs in terms of landslide internal friction angle are improved, and that the groundwater pressure head is another output that would thus be improved.

Recommendation: I cannot recommend this paper for publication in its current form. I think there are numerous details that are absent which makes it impossible to fairly

judge the work the authors have undertaken, and as such it is inappropriate for publication. While the topic itself is of interest and suitable for the journal, there is insufficient information provided to assess whether the results are significant or if the methods are original. Below I include some specific comments that may help the authors to amend this study.

Specific comments:

1. Throughout the paper there is not enough detail provided about the methodology. For example, it is impossible to judge the particle filtering method when the authors say "The assimilation experiment starts on the first day of observation data and occurs daily using the improved PF". This is simply not enough information! Even if the method is published elsewhere, there must be at least a basic explanation of how this process works and the way it was incorporated. It would be impossible for another researcher to replicate almost every aspect of this study based on the information provided, and until that is possible I cannot recommend this for publication. Specific locations where detail is needed: - How were the parameters for TRIGRS obtained? How was hydraulic conductivity measured? - How was rainfall measured? - What is the particle filtering method? - What are the specific soil conditions used to define the three blocks? - How were InSAR measurements corrected? If these methods are original to this study, then every detail of the methods will be necessary to assess the validity; this could be attached in supplementary material if necessary. Where standard or previously published methods have been used, citations are also definitely necessary; I found this study alarmingly lacking in citations.

2. I do not think it is possible to judge the accuracy or validity of the model outputs given the authors' approach, and moreover it is not clear whether the model is over-fitted to the observational data. As far as I can tell, the GPS monitoring data and Sentinel data is used as input to the data assimilation method and as a comparison for the model output. This seems somewhat circular; how can the outputs be independently validated using the same data that is used as input? I can see that the TRIGRS model alone does

not provide outputs close to the observational data, and that the use of DA methods with the observation data gets this closer – but given that the same observational data is used as input and validation seems to me to be circular. It is not clear whether the observations and the assimilation model outputs are similar simply because the DA methods over-fit the model to the observational data, which I would argue is the null hypothesis. As an example, the TRIGRS model outputs data on the groundwater pressure head (phi). The authors could independently validate the model outputs by using field observations of the pressure head as comparison with model predictions; this would test whether over-fitting is an issue.

3. Nowhere in the study or in the supplementary material do the authors ever describe the error margins (levels of uncertainty) attached to their results. It is impossible to assess whether the results are statistically significant given the data presented. I think it is imperative that the authors include error margins on all of their results when revising this study. For example, figures 4,6,7,9,10,12 and 15 should include error bars, and the map figures should at least contain a description of the uncertainty associated with the estimates. I would not consider this study for publication until error margins are fully described.

4. The authors often use jargon that makes it unclear what is being discussed, and I would suggest revising the text to reduce complexity and explain some of the more complicated details. For example, the sentence: "A great number of approaches of DA have been developed in recent years, of which sequential algorithms like particle filtering (PF) are increasingly popular. PF is based on Bayesian theory and originally introduced by Arulampalam into DA (Arulampalam, et al., 2002). It is developed in many DA studies due to the advantage of being unconstrained by state Gaussian distribution and linear assumptions" The terms 'sequential algorithms', 'Bayesian theory', 'state Gaussian distribution and linear assumptions' are not necessarily clear to all readers (this is not, after all, a journal focused on statistical methodology) and should be explained. There are other jargon-y terms used in various places, particularly in the

abstract, where jargon is used extensively, and it should be removed and explained in more detail. Additionally, the authors should not use acronyms before explaining what they are (e.g. 'FS' in the abstract). Finally, there are a number of copy-editing issues that would need to be addressed before final publication. I would be happy to assess and correct these issues in a revised version of the manuscript, but at this stage I think it is inappropriate to list them all given the scale of the other changes required to bring this to a point acceptable for publication.
* * *

---

## Author Comment (AC1) · 16 May 2019

Thanks for your comments. The following is my reply.

General comments: In this paper, a data assimilation experiment was performed using the TRIGRS model to evaluate the stability of landslides. Because the experiment used landslide surface deformation observation data, only areas with FS <1.0 can be calculated. Although the area with FS <1.0 does not necessarily cause slope failure, the stability of the creeping landslide can be inferred based on the trend of the safety factor. If FS changes significantly and lead to deformation rate increases, then this landslide should be focused. However, there is only one equation for parameters and FS, so using FS can estimate only one parameter. Considering soil cohesion and

friction angle are indicators for determining shear strength, we selected one parameter between them. The purpose of this paper is to propose a method to investigate the trend of FS changes through data assimilation with parameter updating. In addition, considering that the observation error is relatively large, it is not accurate to use the observation data of the first day. The part of "Sensitivity analysis" is deleted, instead, the introduction of hydrological data is added.

Specific comments: 1. Thanks. The Data assimilation framework includes background and observations. Background refers to the model used to simulate dynamic processes, and observations can be direct or indirect observation data of the state. In this experiment, "background" is the uncorrected TRIGRS model and its output FS, "observations" are FS calculated from GPS/InSAR monitoring data by equation (1). Related descriptions have been added in the manuscript, pg2, line 5~6. 2. Thanks for your suggestion. The citation and bibliography of this article has been added into the manuscript. 3. This sentence is changed to "It is located at the junction of the Eurasian plate and the Indian Ocean plate. The geological structure is complex and the geological activities are active." 4. Both GPS and InSAR observations are involved in data assimilation calculation. InSAR monitoring points are distributed throughout the landslide surface, but the distribution is uneven and does not coincide with GPS stations. So we put InSAR points and the GPS points into the grid, and some areas where the observation data is sparse are supplemented by interpolation. 5. There is an error in the expression here. In the experiment, the soil and hydrological parameters used in the model remain unchanged, the same as the TRIGRS program running without correction. These parameters were obtained by collecting local geotechnical samples. The rainfall data was obtained by bilinear interpolation from the China Ground Meteorological Information Center (http://data.cma.cn), "China Ground Precipitation $0.5° \times 0.5°$ Grid Data Set (V2.0)". Of course other hydraulic parameters are significant, but as the general comments replies, we can only estimate and update one parameter through one equation. During the experiment, the dynamic change of shear strength needs more attention, which determines the equilibrium state of the soil.

Therefore, we choose the shear strength indicators as parameters for real-time update. The shear strength was chosen as the parameter of interest actually. Cohesion and friction angle are the shear strength indicators. When the cohesion was chosen as the update parameter, the experiment gave similar results. Since the friction angle and the safety factor are nonlinear relationship, it is chosen to propose our method. Internal friction angle is a way to adjust the shear strength in this experiment. Some related explanations have been added to the section 3.2. 6. This part is deleted. 7. Yes. The changing friction angle is effectively correcting for inaccuracies in the hydraulic parameters. Therefore, the sensitivity analysis results are not accurate, so the relevant part is deleted. As is mentioned before, the friction angle is a way to adjust the shear strength in this experiment. It is more meaningful to analyze the correlation between groundwater and rainfall after assimilation. At last, the pressure head and rainfall are displayed in figure 17, and the correlation between them are calculated. 8. The deformation rate maps both calculated from assimilated FS and observations are displayed in figure 15.

Grammar and figures: Thanks for your suggestions. Grammar and figures have been modified.

Please also note the supplement to this comment:
https://www.nat-hazards-earth-syst-sci-discuss.net/nhess-2019-16/nhess-2019-16-AC1-supplement.zip
* * *

---

## Author Comment (AC3) · 16 May 2019

Thanks for your comments. The following is my reply.

Specific comments: 1. The particle filter and its improved algorithm are introduced in detail in another article (Xue, et al., 2018), and this part of the content was added to this manuscript (Part 2). The specific steps of the algorithm can be found in the supplement. The acquisition of hydrological parameters was supplemented in the fourth part of the article. The description of some details in the article was revised, such as landslide blocks and InSAR data. 2. We can't know the true value of the state in the experiment. In fact, the observed data is more credible, and our goal is to use the previously observed data to correct the model's operation. This is to make the model

results not have too much deviation in the future operation. Root mean square error (RMSE) or root mean square difference (RMSD) was used as an indicator to evaluate assimilation results in many studies, such as these articles (Healy and ‐N., 2006, Xie and Zhang, 2010, Zhang, et al., 2013, Bi, et al., 2014). The RMSE/RMSD calculation requires a state true value, or a sample mean instead. With the sample mean calculation, only the fluctuation of the assimilation result itself can be obtained, and the difference between it and the actual value cannot be evaluated. Therefore, it is more appropriate to use the observations for calculation. The data of the groundwater pressure head is not obtained in real time, and the data volume is lacking. As an indirect comparison, we analyzed the correlation between the pressure head and the rainfall sequence of the experimental output in the end of Part 4. 3. The error sequence of GPS observations is displayed separately (Figure 6). In the InSAR observation, we can only know that the overall accuracy of the observation data is better than 1 cm, and cannot obtain the specific value of the observation error of each day. Particle filtering itself is an algorithm that is difficult to perform specific error analysis, usually using RMSE/RMSD as an indicator of the quality of the evaluation results. Therefore, the error bars of the resulting sequence are difficult to draw, and the RMSD sequence (Figure 12) can be used as its quality evaluation. 4. The relevant expression details in the manuscript are modified. The expression of the article has been simplified. The terminology such as "sequential" and "Bayesian theory" has been removed, and the algorithm description of particle filtering has been added in the second part. The use of acronyms has also been revised. Thanks for your suggestion.

References: Bi, H. Y., Ma, J. W., Wang, F. J., 2014. Soil Moisture Estimation Using an Improved Particle Filter Assimilation Algorithm. 2014 Ieee International Geoscience and Remote Sensing Symposium (Igarss). 3770-3773. Healy, S. B., ‐N., T. J., 2006. Assimilation experiments with CHAMP GPS radio occultation measurements. Quarterly Journal of the Royal Meteorological Society 132. 605-623. Xie, X., Zhang, D., 2010. Data assimilation for distributed hydrological catchment modeling via ensemble Kalman filter. Advances in Water Resources 33. 678-690. Xue, C., Nie,

G., Li, H., Wang, J., 2018. Data assimilation with an improved particle filter and its application in the TRIGRS landslide model. Natural Hazards and Earth System Sciences 18. 2801-2807. Zhang, H. J., Qin, S. X., Ma, J. W., You, H. J., 2013. Using Residual Resampling and Sensitivity Analysis to Improve Particle Filter Data Assimilation Accuracy. Ieee Geoscience and Remote Sensing Letters 10. 1404-1408.

Please also note the supplement to this comment:
https://www.nat-hazards-earth-syst-sci-discuss.net/nhess-2019-16/nhess-2019-16-AC3-supplement.zip

---

## Author Comment (AC4) · 23 May 2019

This supplement includes the revised manuscript and data sets.

Please also note the supplement to this comment:
https://www.nat-hazards-earth-syst-sci-discuss.net/nhess-2019-16/nhess-2019-16-AC4-supplement.zip